# Nitrofuran Derivatives Cross-Resistance Evidence—Uropathogenic *Escherichia coli* Nitrofurantoin and Furazidin In Vitro Susceptibility Testing

**DOI:** 10.3390/jcm12165166

**Published:** 2023-08-08

**Authors:** Filip Bielec, Małgorzata Brauncajs, Dorota Pastuszak-Lewandoska

**Affiliations:** 1Department of Microbiology and Laboratory Medical Immunology, Medical University of Lodz, 90-151 Lodz, Poland; malgorzata.brauncajs@umed.lodz.pl (M.B.); dorota.pastuszak-lewandoska@umed.lodz.pl (D.P.-L.); 2Medical Microbiology Laboratory, Central Teaching Hospital of Medical University of Lodz, 92-213 Lodz, Poland

**Keywords:** nitrofuran derivatives, cross-resistance, urinary tract infection, *Escherichia coli*, nitrofurantoin, furazidin, antimicrobials, antimicrobial susceptibility testing

## Abstract

The treatment of urinary tract infections is usually empirical. For example, nitrofuran derivatives, mainly nitrofurantoin (but also furazidin), are used in Eastern Europe. A significant problem is the assessment of the usefulness of furazidin, as there are no standards for susceptibility testing. Additionally, a high percentage of strains resistant to nitrofurantoin should prompt caution when choosing furazidin in therapy. This study aimed to answer the question of whether it is possible to use nitrofurantoin susceptibility for furazidin drug susceptibility analyses and if there is any cross-resistance in the nitrofuran derivatives group. One hundred *E. coli* clinical isolates, obtained from the Central Teaching Hospital of the Medical University of Lodz, were cultured from positive urine samples. For susceptibility testing, microdilution and disk diffusion methods, following EUCAST guidelines, were used. The results showed that the MICs of furazidin were equal to or lower than those of nitrofurantoin in 89% of the tested strains. The MIC_50/90_ values for furazidin were two times lower than those for nitrofurantoin. Positive correlations were found between MICs and growth inhibition zones for both antibiotics. Based on the obtained data and previous studies, it was assumed that the transfer of susceptibility testing results from nitrofurantoin to furazidin is acceptable due to cross-resistance in nitrofuran derivatives.

## 1. Introduction

Urinary tract infections (UTIs) are some of the most common reasons for medical interventions, accounting for about two-fifths of all healthcare-acquired infections (HAI) and one-fifth of community-acquired infections (CAI) [1]. It is estimated that about half of women and one-tenth of men experience an episode of UTI in their lifetime. About one-third of women after the first incident of this disease usually relapse within three months, and almost half relapse within one year [2,3].

Among children, UTIs affect about one-tenth of the pediatric population. UTI in children often accompanies congenital disabilities of the urinary tract, syndromes of metabolic defects, tubulopathies, and immune disorders. In the neonatal period, infections are more common in boys (about four times more often than in girls), and from the age of 1, this trend is reversed [2,3,4].

UTI affects about 10% of men and 20% of women who are elderly. Older age is associated with impairment of general and local defense mechanisms, autonomic nervous system deterioration, menopause, and multiple comorbidities. After the age of 60, the incidence of the disease increases in men (due to urinary outflow disorders due to prostatic hyperplasia). However, older women still suffer from the disease twice as often as older men [2,3,5].

Regardless of the classification of UTIs, their most common etiology are Gram-negative bacteria classified to Enterobacterales and dominated by *Escherichia coli*, especially uropathogenic strains (UPEC, uropathogenic *E. coli*). In community-acquired infections, the share of *E. coli* as the etiological factor is over 50%, regardless of the examination [1,6,7,8,9,10,11,12]. On the other hand, this percentage falls below half of all urine-cultured pathogens in healthcare-acquired UTIs. The share of other Gram-negative bacilli from the non-fermenting group (incl. *Pseudomonas* spp., *Acinetobacter* spp.) is significantly increasing [1,11,12,13]. Gram-positive bacteria are much less likely to be the etiological factor of UTI–bacteria of the genus *Enterococcus* are most often identified, the share of which in HAI is more than twice as high as in CAI. However, the hypothesis requires reflection on whether such a large share of *Enterococcus* spp. in the tested urine samples is not falsely inflated in some studies by the use of boronic acid to preserve urine samples for microbiological tests—in vitro studies have shown that *Enterococcus* spp. is resistant to the inhibitory effects of this substance [14]. *Staphylococcus aureus* and other coagulase-negative staphylococci (with the predominance of *Staphylococcus saprophyticus*) are also visible in the statistics of the etiology of UTI. A summary of the most common etiology of UTI depending on the source of infection (CAI vs. HAI) is presented in Table 1 [1,6,7,8,9,10,11,12,13,15,16,17,18]. Some analyses [6,18,19,20] also indicated a significant percentage (1–11%) of *Streptococcus agalactiae* among the Gram-positive etiological factors of UTI; however, due to limited reports (and a lack of information on healthcare or community sources), they are not included in Table 1.

In children, *E. coli* is also the most common etiological agent of UTI, present in more than half of all cases, regardless of the presence of risk factors for UTI [2,3,21].

UTI treatment is usually empirical. The choice of an antibiotic should consider drug bioavailability, tolerability and side effects, drug sensitivity, antibacterial spectrum, local data on drug susceptibility of uropathogens, efficacy confirmed by clinical trials, and therapy costs. Current guidelines of European scientific societies recommend nitrofuran derivatives, fosfomycin, or β-lactams as the first-line treatment of uncomplicated UTI. From the group of nitrofuran derivatives, international guidelines recommend nitrofurantoin, but locally, where these drugs are available in Europe, they also recommend furazidin (Poland, Latvia, North Macedonia, Albania, Kosovo, Ukraine, Belarus, and Russia) or nifurtoinol (Belgium and Italy) [22,23,24]. Nitrofuran derivatives, which reach therapeutic concentrations only in the urine, should be preferred in therapy due to the need to protect antibiotics. They are not recommended for treating complicated UTIs because they only marginally penetrate tissues that are usually infected in these cases [25,26].

Although the etiology of UTI has mostly stayed the same for many years, modern bacterial pathogens have acquired several new features that make the effective treatment of UTI more and more difficult. The most important of these features is the acquisition of many antibiotic resistance mechanisms, including nitrofuran derivatives. This phenomenon applies to both healthcare- and community-acquired strains [2,27]. In recent years, there has been an increased use of antibiotics to treat UTIs in outpatient clinics, including nitrofuran derivatives [28,29].

In the 21st century, numerous epidemiological studies of drug susceptibility of the most common uropathogens were conducted. Figure 1 summarizes the resistance rate to nitrofurantoin among strains of *E. coli* causing UTI. The results come from studies performed in different regions of the world, in different periods, and in groups of different sizes [2,7,8,10,13,15,16,17,18,19,20,21,30,31,32,33,34,35,36,37,38,39,40,41,42]. In most cases, the percentage of resistance to nitrofurantoin was low and often did not exceed 3%. However, in some cases, the observed resistance exceeded 20% (Cameroon, 2014; Ethiopia 2015–16; Pakistan 2019–20; India, 2016) [15,19,21,38] and, notably, even exceeded 30% in Poland, 2013 [8].

The rapid implementation of appropriate empirical therapy in suspected UTI is even more critical in the era of increasing antimicrobial resistance due to the possibility of the disease evolving into urosepsis, the treatment of which may be complicated by multidrug resistant organisms—the incidence of which in urosepsis is increasing [43].

A significant problem in the selection of the drug in the empirical treatment of UTI in many European and world countries is the lack of current data on local drug susceptibility (rarely conducted analyses, drug susceptibility determined in different ways, different study groups, different criteria for the diagnosis of UTI). An additional problem—specific mainly to Eastern European countries—is the assessment of the usefulness of furazidin, which is available in Poland or Ukraine without a prescription. Firstly, the problem is the inability to test drug susceptibility in routine testing with any standard to compare. There are no tests for in vitro diagnostics of susceptibility to this antimicrobial drug (the results are unreasonably extrapolated from nitrofurantoin susceptibility test) [23]. Secondly, disturbing reports of a high percentage of strains resistant to nitrofurantoin, even exceeding 30%, should prompt caution when choosing furazidin in therapy [8,15,19,21,38].

This study aimed to check the correlation between susceptibility testing of *E. coli* strains isolated from urine to furazidin and nitrofurantoin. A good basis may be the publication by Klesiewicz et al. [44], who have suggested the compatibility of antimicrobial susceptibility for both nitrofuran derivatives, tested against *E. coli* and *S. aureus* strains. However, the small study group requires confirmation, as only 18 clinical *E. coli* isolates have been analyzed, which is not a statistically significant sample size. We also attempted to answer the question of whether it is possible to use nitrofurantoin susceptibility tests for local drug susceptibility analyses and planning targeted furazidin therapy due to the lack of appropriate standardized diagnostic tests for furazidin susceptibility testing.

## 2. Materials and Methods

The study group consisted of 100 *E. coli* clinical isolates cultured from positive urine samples obtained between February and August 2021 from the Medical Microbiology Laboratory at the Central Teaching Hospital of the Medical University of Lodz. In order to increase the uniqueness of the tested isolates and to avoid repetition, it was agreed upon with the laboratory that only *E. coli* strains from the first positive urine cultures of specific patients will be secured for this research. All bacteria were stored in Viabank^TM^ storage beads (Medical Wire & Equipment, Corsham, UK) at −80 °C maximum and regenerated on Columbia Agar with 5% sheep blood (Thermo Fisher Scientific, Waltham, MA, USA), 18 ± 2 h at 35 ± 1 °C in atmospheric conditions.

The susceptibility for nitrofurantoin and furazidin was tested using broth microdilution and disk diffusion methods. All determinations were made in triplicate.

The broth microdilution susceptibility test was performed following ISO 20776-1:2019 [45]. The bacteria were inoculated on 96-well titer plates in a series of two-fold dilutions of antimicrobials (256–0.5 mg/L) in Mueller–Hinton broth (Thermo Fisher Scientific, Waltham, MA, USA) and incubated sealed for 18 ± 2 h at 35 ± 1 °C in atmospheric conditions. MIC was defined as the concentration demonstrating a lack of growth according to the European Committee on Antimicrobial Susceptibility Testing (EUCAST) reading guide for broth microdilution version 3.0 [46].

The disk diffusion susceptibility test was performed following the EUCAST methodology, ver. 9 manual [47]. The discs with antimicrobials were applied to the surface of the inoculated Mueller–Hinton agar plates (Thermo Fisher Scientific, Waltham, MA, USA) and incubated for 18 ± 2 h at 35 ± 1 °C in atmospheric conditions. The discs with nitrofurantoin (100 μg) were obtained from the manufacturer (Liofilchem, Roseto degli Abruzzi, Italy). The discs with furazidin (200 μg) were prepared earlier the same day—blank discs (Thermo Fisher Scientific, Waltham, MA, USA) were soaked with furazidin solution. The inhibition zone diameter was measured manually with a caliper according to the EUCAST reading guide for the disk diffusion method [48].

Stock solutions of nitrofurantoin and furazidin were prepared using pure substance powders obtained from the manufacturer (Selleck Chemicals, Houston, TX, USA). It has been proven before that such a source is the best for quantitative assays [49]. The agents were first dissolved in DMSO (Thermo Fisher Scientific, Waltham, MA, USA) and then diluted with distilled water. Ultimately, the 5120 mg/L stock solutions had a DMSO concentration of ~10%. The broth microdilution susceptibility test with DMSO showed that this concentration did not inhibit the growth of the tested bacteria.

Statistical analysis was performed using Statistica 13 software (TIBCO Software Inc., Palo Alto, CA, USA). The distribution of collected data was checked using the Shapiro–Wilk test. All variables were distributed non-normally. The correlations were checked using Spearman’s test. A *p*-value of 0.05 was considered the limit of statistical significance.

### Ethical Issues

The presented study was conducted with high ethical standards in accordance with the Declaration of Helsinki. The study involved only anonymized records, without the possibility of identifying a specific human being. All bacterial strains were previously secured in the culture collection of our research unit, using consecutive code identification numbers. The only clinical data concerned the type of biological material from which the bacterial strain was isolated. The research plan was approved by the Ethics Committee of the Medical University of Lodz (protocol code RNN/50/21/KE of 9th February 2021).

## 3. Results

Interpretation of the susceptibility to nitrofurantoin of the tested strains according to EUCAST 2023 [50] showed a resistance rate of 13% for the broth microdilution method (MIC breakpoint = 64 mg/L) and 11% for the disc diffusion method (zone diameter breakpoint = 11 mm). For the 2 *E. coli* isolates where the interpretation did not comply, the MICs were 128 mg/L for both, and the inhibition zones were 11 and 13 mm.

MICs and zones of inhibition negatively correlated with each other for both nitrofurantoin (ρ = −0.73, *p* < 0.001) and furazidin (ρ = −0.79, *p* <0.001). Figure 2 shows the MIC and zone diameter distribution histograms.

Comparing the determinations for both tested nitrofuran derivatives, the MICs of furazidin were equal to or lower than that of nitrofurantoin in 89% of the tested strains. For the remaining 11 isolates, the furazidin MICs were 1 two-fold dilution higher than nitrofurantoin. In total, nitrofurantoin MIC_50/90_ = 16/128 mg/L, and for furazidin MIC_50/90_ = 8/64 mg/L.

There were positive correlations found between the MICs of both antibiotics (ρ = 0.67, *p* < 0.001), as well as between their zones of inhibition (ρ = 0.81, *p* < 0.001).

The obtained raw results of susceptibility testing are presented in the Appendix A.

## 4. Discussion

This study aimed to answer the question of whether it is possible to extrapolate nitrofurantoin susceptibility testing results for another nitrofuran derivative, furazidin. To check if there is a cross-resistance between these antimicrobials, we simultaneously assessed MICs and growth inhibition zones for nitrofurantoin and furazidin on a statistically large group of clinical *E. coli* isolates.

The obtained results indicate the occurrence of cross-resistance between nitrofurantoin and furazidin for *E. coli* bacteria. In almost nine-tenths of cases, the MIC values obtained for furazidin were lower than those for nitrofurantoin. Comparing the MIC_50_ and MIC_90_ values, the obtained values were two times lower for furazidin than for nitrofurantoin. Similar results were obtained by Klesiewicz et al. [44] on a group of 18 clinical isolates of *E. coli*. Comparable results obtained in two independent research centers only confirm their veracity.

Another publication supporting the obtained results is the study of Mannisto and Karttunen [26], who were the first to describe the pharmacological properties of furazidin in 1979. They also compared the antibacterial activity of nitrofurantoin and furazidin. The results of all three published experiments showed a higher activity of furazidin, expressed as lower MICs, which suggests that furazidin may be a valuable alternative to nitrofurantoin. Perhaps it would be worth considering the introduction of this antimicrobial through central registration throughout the European Union.

The proposed thesis on cross-resistance between furazidin and nitrofurantoin is also supported by the fact that there are statistically significant positive correlations between the results of susceptibility testing with both recognized methods—broth microdilution and disc diffusion.

The issue of nitrofuran derivatives cross-resistance has already been raised in the past. In 1952, Paul et al. [51], based on the study of cross-resistance of *E. coli* to various compounds of nitrofuran derivatives, observed the occurrence of cross-resistance, but encountered some limitations. The tested molecules were divided into two classes. *E. coli* bacteria resistant to a Class I substance were reciprocally cross-resistant to all members of this class but remained susceptible to nitrofuran derivatives of the II class. The II class differed from the I class in chemical structure by having a carbon atom between the carbonyl group and the terminal group of the nitrofuran sidechain.

The study by Borkowska-Opacka et al. [52] also found the cross-resistance of *E. coli* bacteria to nitrofuran derivatives—furazolidone, nitrofurazone, and nitrofurantoin. Due to their chemical structure, these substances could be included in class I according to Paul et al. [51]. The furazidin we studied could also be classified as class I. Nitrofurantoin and furazidin chemically differ only in the hydrocarbon chain length connecting the functional groups at the ends (see Figure 3).

Another antibiotic from the group of nitrofuran derivatives used in urinary tract infections is nifurtoinol, which has similar pharmacological properties to nitrofurantoin and furazidin [53]. The chemical structure of nifurtoinol (see Figure 3) allows it to be included in class I according to Paul et al. [51]. It may be therefore concluded that the cross-resistance described in this article also applies to nifurtoinol. It would be interesting to test this thesis experimentally.

Moreover, yet another observation made during the study is worth mentioning because of its therapeutic significance. In 2% of the isolates tested, the clinical interpretation of nitrofurantoin susceptibility was inconsistent depending on the method used. Whereas the results of broth microdilution (reference method) classified a given strain as “resistant”, the results of disc diffusion method classified it as “susceptible”. This is a very major error that can lead to poor clinical decisions. It is suggested that EUCAST revise its nitrofurantoin breakpoints [50]. In this case, the comparison with the US Clinical and Laboratory Standards Institute (CLSI) breakpoints does not make sense, because this organization recommends using a different concentration of nitrofurantoin in the disc [54].

### Study Strengths and Limitations

The main strength of the study was the statistically large group of clinical isolates tested, which ensured the appropriate power of the tests used and allowed for analysis and interpretation with a degree of probability bordering on certainty. Another advantage is the use of two widely recognized methods of drug susceptibility testing.

The limitation of the study was that only two antimicrobials from the group of nitrofuran derivatives were tested with the designed methodology. Undoubtedly, it would be valuable to include nifurtoinol in the experiment, but it was not possible to obtain it from any of the reagent distributors available to our research unit.

## 5. Conclusions

Our study showed significant correlations between the results of antimicrobial susceptibility tests for nitrofurantoin and furazidin—the two antimicrobials from the group of nitrofuran derivatives with a similar chemical structure. We also confirmed the higher antibacterial activity of furazidin compared to nitrofurantoin.

Summarizing the presented evidence, the transfer of susceptibility testing results from nitrofurantoin to furazidin is acceptable due to cross-resistance in nitrofuran derivatives. This information is crucial for developing local guidelines for empirical antibiotic treatment of UTIs in countries with limited or no availability of nitrofurantoin.

Since all previous cross-resistance studies of nitrofuran derivatives were carried out mainly on *E. coli* bacteria, confirming the observations for other species potentially showing susceptibility to nitrofuran derivatives, e.g., *Enterococcus faecalis* or *Staphylococcus saprophyticus*, is necessary. In addition, it would be worth repeating the experiment using another nitrofuran derivative—nifurtoinol.

## Figures and Tables

**Figure 1 jcm-12-05166-f001:**
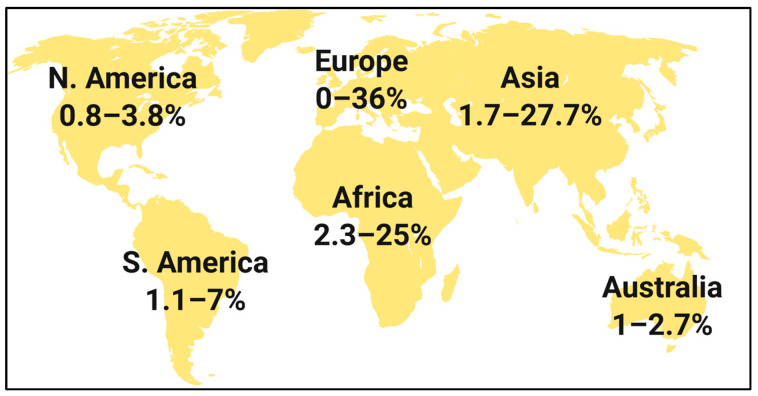
Overview of data on *Escherichia coli* urine isolates nitrofurantoin resistance in the 21st century worldwide reports [2,7,8,10,13,15,16,17,18,19,20,21,30,31,32,33,34,35,36,37,38,39,40,41,42].

**Figure 2 jcm-12-05166-f002:**
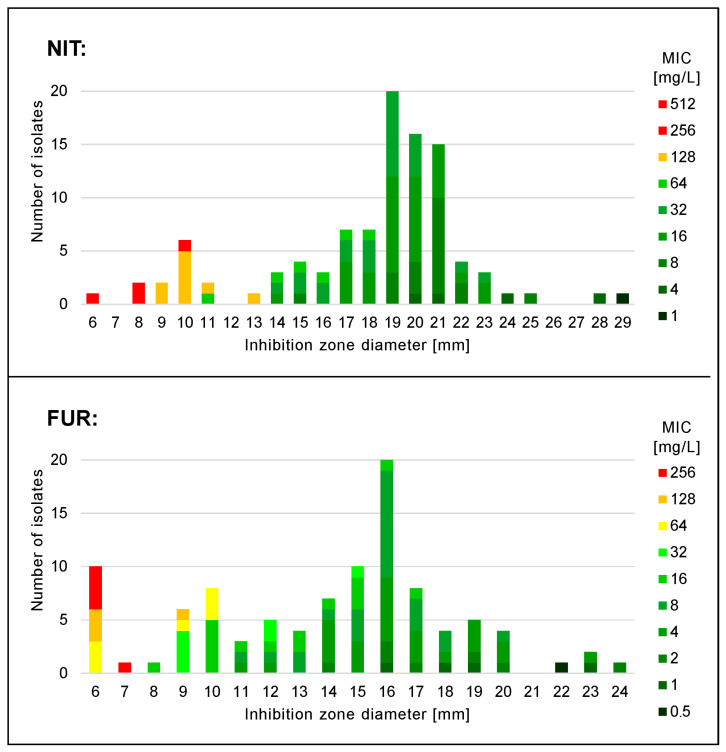
Nitrofurantoin (NIT) and furazidin (FUR) minimal inhibitory concentration (MIC) and zone diameter distribution histograms.

**Figure 3 jcm-12-05166-f003:**
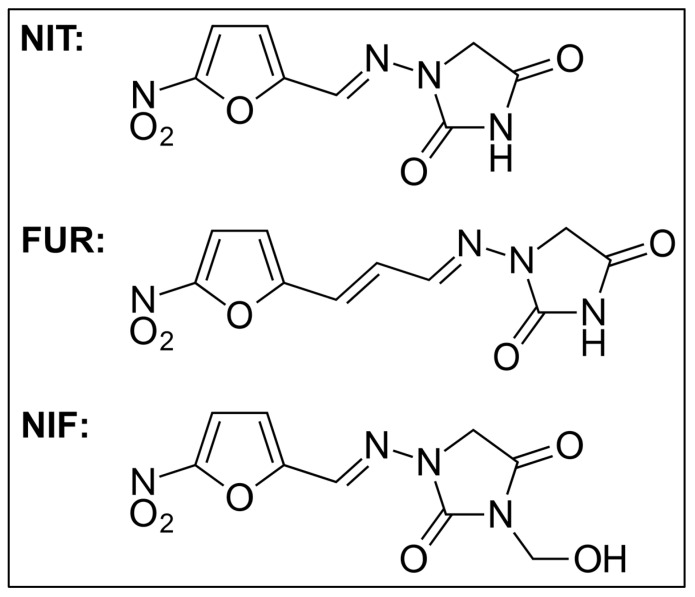
Structural chemical formulas of nitrofurantoin (NIT), furazidin (FUR), and nifurtoinol (NIF).

**Table 1 jcm-12-05166-t001:** Summary of the most common urinary tract infection (UTI) bacterial etiology depending on the source of infection community-acquired (CA-UTI) vs. healthcare-acquired (HA-UTI) [1,6,7,8,9,10,11,12,13,15,16,17,18].

Bacteria		CA-UTI	HA-UTI
Gram-negative	*Escherichia coli*	51–84%	25–45%
	*Klebsiella pneumoniae*	4–17%	10–38%
	CESP group	2–9%	4–11%
	Non-fermenters	0–7%	8–19%
Gram-positive	*Enterococcus* spp.	2–16%	4–42%
	*Staphylococcus* spp.	1–8%	3–6%

CESP group = *Citrobacter* spp., *Enterobacter* spp., *Serratia* spp., *Proteus* spp., *Providencia* spp., *Morganella* spp.; Non-fermenters = *Pseudomonas* spp., *Acinetobacter* spp.

## Data Availability

The data presented in this study are available in Appendix A.

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
