# Peer review of "Nitrofuran Derivatives Cross-Resistance Evidence—Uropathogenic Escherichia coli Nitrofurantoin and Furazidin In Vitro Susceptibility Testing"

_jcm, 2023, doi:10.3390/jcm12165166_

Round 1

Reviewer 1 Report

Nitrofuran derivatives cross-resistance evidence – uropathogenic Escherichia coli nitrofurantoin and furazidin in vitro susceptibility testing, an article by Filip Bielec et al aimed to answer whether it is possible to use nitrofurantoin susceptibility for furazidin drug susceptibility analyses. To do so, the authors check if there is cross-resistance between these antimicrobials, and assessed MICs and growth inhibition zones for nitrofurantoin and furazidin on a statistically large group of clinical E. coli isolates.

How did the authors rule out that the 100 E.coli isolates tested were different strains and were not repetitions?

However, the major concern is that the question asked is in itself not compelling and does not contribute anything interesting or novel. The authors themselves discuss that cross resistance between the two derivatives have been reported earlier. However they do not cite another very similar article, from their country itself; ‘COMPARATIVE IN VITRO STUDIES OF FURAZIDIN AND NITROFURANTOIN ACTIVITIES AGAINST COMMON UROPATHOGENS INCLUDING MULTIDRUG-RESISTANT STRAINS OF E. COLI AND S. AUREUS’  by Karolina Klesiewicz et. al. published in 2018.

There is no new knowledge addition by this work. What is known already, has yet again been presented.

English editing is required and at places, sentences are left incomplete. Some examples are listed below:

Line 13 Abstract: the sentence looks incomplete.  “…………but in Eastern Europe also furazidin.”

Line 17 Abstract: ‘if there is’ …instead of…… ‘if is there any cross-resistance’

Line 21: Comparing the results? Instead of Comparing the determinations

Line 75: Too long and confusing sentence - “Current guidelines of European scientific societies recommend….”

Line 79: Not clear- “Nitrofuran derivatives which reach therapeutic concentrations only in the urine and should be preferred…”

Line 112: ‘in vitro’ italize, also to be done at other places

English editing is required and at places, sentences are left incomplete. Some examples are listed below:

Line 13 Abstract: the sentence looks incomplete.  “…………but in Eastern Europe also furazidin.”

Line 17 Abstract: ‘if there is’ …instead of…… ‘if is there any cross-resistance’

Line 21: Comparing the results? Instead of Comparing the determinations

Line 75: Too long and confusing sentence - “Current guidelines of European scientific societies recommend….”

Line 79: Not clear- “Nitrofuran derivatives which reach therapeutic concentrations only in the urine and should be preferred…”

Line 112: ‘in vitro’ italize, also to be done at other places

Author Response

We kindly thank you for all the comments and tips received. Thanks to them, it was possible to improve the manuscript, which will increase the quality of publication.

Our great oversight was the failure to include Klesiewicz et al. We did not find it either at the stage of designing the study or writing the article, which was probably due to poor indexing of the journal Acta Poloniae Pharmaceutica in the scientific publications databases. Thanks to a reviewer, this paper is now cited and reinforces the strength of its conclusions.

We were surprised by the comments about the quality of the English language. The work was checked by a translator before being sent. After the review, the work was checked again by the translator and the web application Grammarly. We hope it now meets the strictest editorial standards.

In addition, other corrections were made to the manuscript as suggested by other reviewers.

Reviewer 2 Report

The current article (jcm-2522354) highlighted that furazidin susceptibility testing could be used an alternative to nitrofurantoin susceptibility testing for E. coli involved in UTI. The authors claimed that present study finding could helpful in developing antibiotic treatment regime for UTIs in countries with limited or no availability of nitrofurantoin.

Similar findings have already been reported in literature, so the study has limited novelty in terms of clinical applications. Furthermore, the data presented in the article is quite limited,  I will suggest to consider this article in the category of short communication instead of Full Research article.

The following queries and comments may be clarified:

Comments:

  • Abstract: The methodology should be scientifically revised including the sampling area/hospital. The results in abstract section should be described in more quantitative manner.
  • Introduction is too lengthy. Table 1 and 2 may be deleted.
  • Figure 1: Number of observation may be changed to Number of isolates.
  • Figure 2 and Figure 3 seems repetition of data from Figure 1.
  • Figure 4 may be deleted.
  • Conclusion should be revised as statements given in the conclusion are much generalized. So it must be based on the current study’s findings.

Minor editing of English language required

Author Response

We kindly thank you for all the comments and tips received. Thanks to them, it was possible to improve the manuscript, which will increase the quality of publication.

In the “Abstract” - corrections were made in accordance with the reviewer's suggestions.

In the “Introduction” - Table 1 has been, we believe, important to show why strains of E. coli (UTI most common etiological agent) were tested. Table 2 converted to a figure (now Figure 1) - to show the scale of resistance to nitrofurantoin, the figure should be clearer for reader.

In the “Results” – Figure 1 (now Figure 2) has been corrected as suggested by the reviewer. Figures 2 and 3 have been removed and correlation statistics are now included in the text.

In the “Discussion” - Figure 4 (now Figure 3) stayed, the chemical formula of nifurtoinol has been added to it. We believe that this figure should remain, because the recipient, while reading, will most likely want to visually compare the structural formulas of the described substances.

In the “Conclusions” - revised as suggested by the reviewer.

In addition, other corrections were made to the manuscript as suggested by other reviewers.

Round 2

Reviewer 1 Report

accept

Reviewer 2 Report

Overall the manuscript has been significantly improved in terms of scientific and technical quality. Most of the comments have been addressed.